# A Genomic Evaluation of Six Selected Inbred Lines of the Naturalized Plants of Milk Thistle (*Silybum marianum* L. Gaertn.) in Korea

**DOI:** 10.3390/plants12142702

**Published:** 2023-07-20

**Authors:** Jeehyoung Shim, Su Young Hong, Jae-Hyuk Han, Yeisoo Yu, Eunae Yoo, Jungsook Sung, Joong Hyoun Chin, O New Lee

**Affiliations:** 1Department of Food and Nutrition, Chung Ang University, Seodong-daero 4726, Daedeok-myeon, Anseong 17546, Republic of Korea; gyoung.feb@gmail.com; 2EL&I Co., Ltd., Hwaseong 18278, Republic of Korea; 3Genomics Division, National Institute of Agricultural Sciences, Rural Development Administration, Jeonju 54874, Republic of Korea; suyoung@korea.kr; 4Food Crops Molecular Breeding Laboratory, Department of Integrative Biological Sciences and Industry, Sejong University, Seoul 05006, Republic of Korea; 0724jh@gmail.com; 5DNACare Co., Ltd., Seoul 06730, Republic of Korea; yeisooyu@dnacare.co.kr; 6National Agrobiodiversity Center, National Institute of Agricultural Sciences, Rural Development Administration, Jeonju 54874, Republic of Korea; eung77@korea.kr (E.Y.); sjs7861@korea.kr (J.S.); 7Convergence Research Center for Natural Products, Sejong University, Seoul 05006, Republic of Korea; 8Department of Bioindustry and Bioresource Engineering, Sejong University, Seoul 05006, Republic of Korea

**Keywords:** *Silybum marianum*, milk thistle, InDel, resequencing, naturalized plants

## Abstract

Milk thistle (*Silybum marianum*) belongs to the Asteraceae family and is a medicinal plant native to the Mediterranean Basin. Silymarin in achene is a widely used herbal product for chronic liver disease. There is growing interest in natural medicine using milk thistle in Korea, but the raw material completely relies on imports. Despite its economic importance, phenotypic evaluations of native resources of milk thistle in Korea have not been carried out. In addition, genomic research and molecular marker development are very limited in milk thistle. In this study, we evaluated 220 milk thistle resources consisting of 172 accessions collected from the domestic market, and 48 accessions isolated from 6 accessions distributed by the National Agrobiodiversity Center in Korea. Six plant characteristics (height, seed weight, number of flowers, seed weight per flower, spine length, and color at harvest) were measured, and six samples (M01–M06) were selected to represent the genetic diversity of the population for genomic research. To develop PCR-based and co-dominant insertion/deletion (InDel) markers, we performed genome-wide InDel detection by comparing the whole-genome resequencing data of the six selected accessions with the reference genome sequence (GCA_001541825). As a result, 177 InDel markers with high distinguishability and reproducibility were selected from the 30,845 InDel variants. Unknowingly imported alien plant resources could easily be genetically mixed, and jeopardized seed purity can cause continuous difficulties in the development of high value-added agricultural platforms utilizing natural products. The selected plant materials and 177 validated InDel markers developed via whole-genome resequencing analysis could be valuable resources for breeding, conservation, and ecological studies of natives to Korea, along with acceleration of *Silybum marianum* industrialization.

## 1. Introduction

Milk thistle (*Silybum marianum* L. Gaertn.) is an annual-to-biennial plant of the Asteraceae family, native to the Mediterranean area, and is now a widely cultivated officinal plant [1,2,3]. It is a diploid species (2n = 34) and an autogamous plant with an average outcrossing rate of 2% under field conditions [4,5,6]. Milk thistle has a glossy, brown–black-to-greyish husk achene with cypselae. In the dry pericarp and seed coat of the achene, flavonolignans (about 70–80%), as well as polymeric and oxidized polyphenolic compounds consisting of a mixture of flavonoids, are accumulated [1,3]. The health-promoting properties of milk thistle are attributable to flavonolignans, commonly referred to as silymarin [2,7]. Silymarin content ranges from 1.5% to 3.0% of the achene dry weight, but may exceed 4.0% [7]. Silymarin is composed of the six representatives of flavonolignans: silybin, isosilybin, silychrstin, isosilychristin, silydianin, and silimonin [3]. The main component of silymarin is silybin (C_25_H_22_O_30_, molecular weight of 482.441), which is a mixture of two diastereomers: silybins A and B. Silybin has pharmacologically relevant actions for human liver diseases (e.g., liver-regenerating properties, anti-inflammatory, immunomodulating, antifibrotic, and antioxidating effects), as well as the clinical potential in patients with viral hepatitis, drug-induced liver injury, and non-alcoholic fatty liver disease [1]. Aside from silymarin, milk thistle achenes have a high oil content (20–30%) [2,4]. Byproducts generated from silymarin extraction and other fractions of biomass can be used in various ways, namely edible oil, fodder, cosmetics, and bioenergy production [4].

Milk thistle is not only rich in nutrients and offers medicinal benefits, but is also well-adapted to suboptimal growing conditions [8]. It is considered one of the most interesting alternative crops in the Mediterranean environment [9,10] and has been tested as a potential commercial seed crop in Canada and New Zealand [4,11,12]. In North America, commercial cultivation has recently become more significant due to the growing popularity of herbal supplements and increasing demand for milk thistle extract from the pharmaceutical industry [13]. In Korea, milk thistle is recognized as a naturalized plant introduced by artificial or natural methods that is able to reproduce and survive in the wild [14]. Milk thistle extract was ranked 10th, with a yield of 45.1 billion KWN (34 million USD) in the health functional food market, itself estimated to have a value of 4.03 trillion KWN (2.8 billion USD) (Ministry of Food and Drug Safety of Korea, 2021). On the other hand, domestic pharmaceutical companies rely on imported materials of milk thistle from foreign countries, such as Poland, France, the U.S., India, etc. In order to develop a sustainable milk thistle industry in Korea, it is necessary to evaluate domestic resources and develop domestic milk thistle varieties. 

DNA-based molecular markers are a valuable tool in both basic and applied research, such as fingerprinting genotypes, analyzing genetic diversity, and marker-assisted breeding [15,16,17,18]. Phenotypic characters are generally influenced by the environmental factors and developmental stage of the plant, and agronomic practices. In contrast, molecular markers based on DNA sequence polymorphisms are independent of environmental factors and show high polymorphism, reproducibility, and reliable identification. NGS has allowed us to identify a massive number of single-nucleotide polymorphisms (SNPs) and insertion and deletion (InDel) polymorphisms between highly homologous genomes [19]. InDel markers are relatively easy to genotype based on their fragment-length polymorphism, without special infrastructure to perform SNP genotyping, and extensively cover the whole genome [15,20,21]. InDels are the second most abundant forms of genetic variation in plants and humans, next to SNPs [21,22,23]. InDel markers offer advantages in their multiallelic nature and codominant inheritance. InDels generally have a low frequency of homoplasy, which represents the probability of two InDel mutations of exactly the same length occurring at the same genomic location, allowing InDels to be confidently related to identity-by-descent [19]. For this reason, the usefulness of InDels over SSR markers was demonstrated in analyzing the interspecific structure of cultivated citrus genetic diversity. InDels are genetic variants that can have a more significant impact on protein structure and function than single-base changes, thus allowing for their use in the development of phylogenetic markers [24]. InDel markers have been developed in various crop species, such as rice, soybean, hot pepper, and maize [25,26,27,28,29]. However, in milk thistle, genome-wide PCR markers have not yet been reported. There have been some recent attempts to create shatter-resistant mutant lines using DArT (diversity array technology), and analyze genetic diversity using the SCoT (start codon-targeted) marker system [4,30]. By harnessing the first draft of the whole genome of *S. marianum*, it is now possible to detect genome-wide InDel polymorphisms among different accessions using whole-genome resequencing to guide the efficient development of PCR-based markers [18].

In this study, six accessions that showed significant differences in agricultural characteristics were selected to be used as initial materials for genomic research of milk thistle. We used resequencing data from the six accessions in comparison with the reference genome sequence (assembly ID: GCA_001541825.1) to identify 30,845 polymorphisms across the genome. Furthermore, we converted 3410 InDel polymorphisms with a separation ratio of 3:3 between the reference allele and the alternative allele among the six re-sequenced accessions to PCR-based InDel assay markers. After experimental validation, 177 InDel markers with high distinguishability and reproducibility were selected. These plant materials and InDels will be useful resources for genetic research and breeding programs of *S. marianum*.

## 2. Results

### 2.1. Phenotypic Evaluation of 220 Milk Thistle Accessions 

All analyzed traits showed a continuous unimodal distribution among the 220 milk thistle accessions, except for plant color at harvest (Figure 1). The plant height ranged from 32 cm to 176 cm, with an average of 109 cm, when erect and branched in the upper part of the plant (Table 1, Figure 2). The basal leaves were large and glabrous with spiny margins. Each stem ended in a flower with a spiny bract, and the average number of flower heads was 37.5 per plant. The average seed weight per plant in 220 accessions was 112.2 g, and ranged from 0.2 g to 311.4 g. Phenotypic comparisons between M01–06 and 220 accessions showed no significant differences in SW, FHN, and PC. However, there were significant differences in PH, SW/FHN, and SL, as the value of six accessions were higher than 220 accessions (Table 1). All traits were highly correlated (*p* ≤ 0.01) with each other, except for the number of flower heads and spine length (Appendix A). Positive correlations were observed among the traits, while spine length showed negative correlations with all the other traits. Among the six traits, two traits (spine length and plant color at harvest) showed a low correlation coefficient and high *p*-value with the other traits (plant height, seed weight, flower head number, and seed weight/flower head number). Although the *p*-value was low and significant, it is likely that spine length and plant color had a low correlation with the other traits. 

To assess the phenotypic diversity of 220 accessions, six agronomic traits were evaluated and analyzed using principal component analysis (PCA) and cluster analysis. The first two PCs accounted for approximately 74%. PC1 represented the “crop yield” such as seed weight (r = 0.97), flower head number (r = 0.91), plant height (r = 0.86), and seed weight/flower head number (r = 0.77), which accounted for 56.2% of total variance (Table 2). PC2 represented the “appearance of plants at harvest”, such as spine length (r = 0.84) and plant color at harvest (r = −0.72), which accounted for 17.6% of total variance. Using a hierarchical cluster analysis based on the six agronomic traits, a dendrogram classified the 220 milk thistle accessions into three main groups (Appendix A). We selected six representative milk thistle accessions (M01–M06) as initial materials for genomic research, including two Korean natives (M05 and 06). M01, M03 and M04, and M02, M05, and M06 were each assigned to one of three clusters. After the selection, we re-evaluated the six representative plants for six agricultural characteristics from 2019 to 2021 (Appendix A). The six accessions showed significant differences in all six traits. Plant height and involucre diameter were the highest in M04 among the six accessions, whereas the number of flower heads, 100-achene weight, and spine length were the highest in M05. Plant height was the highest in M04 (124.3 cm and 55.8 mm, respectively), but was the smallest in M03 (95.3 cm and 47.1 mm, respectively). The number of branches of M06 was the highest (23.4) among the six accessions.

### 2.2. Sequencing and Mapping Summary 

To develop genome-wide DNA markers in *S. marianum*, we produced 120.5 Gb of raw sequence across a total of six accessions, which ranged from 16.7 Gb (M01) to 22.7 Gb (M03) and about 20.1 Gb for each accession on average (Appendix A). After LQ and adapter sequences were trimmed using “fastp”, with a Q-score ≥ 20 and a minimum length ≥ 36 bp, the trimmed sequence remained 117 Gb in total (97.1% of raw data). It ranged from 16.3 Gb (M01) to 21.9 Gb (M03, M05), and 19.5 Gb per accession on average.

In total, 687.7 million trimmed reads were mapped to the *S. marianum* draft assembly using BWA-mem, which showed an 88.2% average mapping rate (Table 3). On average, 114.6 million read/samples were mapped. The mapping rate per each accession ranged from 85.3% (M02) to 90.0% (M01, M03). About 92.3 million reads were unmapped (11.8%). About 83.6% of the mapped reads were properly paired, accounting for 81.9 million to 109.9 million reads for each accession.

### 2.3. Identification and Validation of Genome-Wide InDels

Although both insertions and deletions (InDels) and single-nucleotide polymorphisms (SNPs) were identified in this study, we focused on InDels for further analysis and discussion. In total, 3,518,667 raw InDels were called via GATK haplotypeCaller. After variant filtering, 238,988 InDel variants remained. In addition, after the strict filtering of heterozygotes for the reference allele, 30,845 InDels were grouped by modified allele frequency (Appendix A, Table 4). Of these, 3410 InDels exhibited three homozygotes for the reference allele and three homozygotes for the alternative allele, which refers to group ‘303’ polymorphic type. To select indel markers that were highly discriminatory and easy to use for profiling the genotypes of six resources, only the genotype group ‘303′ was used for InDels marker development in this study. Of them, 234 InDels with a size difference of more than 15 bp between the reference allele and alternative allele were selected for marker design (Table 4). Length differences less than 39 bp accounted for 82% of InDels. Three InDels with length differences of more than 100 bp (107 bp, 176 bp, and 248 bp) were deleted in this study. Therefore, a total of 231 InDels were finally used, consisting of 107 (46.3%) inserted InDels and 124 (53.7%) deleted InDels. All InDels were >200 bp inward from the beginning or end of a contig, comprising sufficient sequences available for primer design. The InDel size difference generated by the tested primers ranged from 30 to 217 bp. Of them, 177 (77%) InDels yielded a single PCR fragment and showed polymorphism, 33 (14%) did not amplify a product, and 21 (9%) showed no polymorphism (Appendix A). Only 10 InDels showed a ‘303’ allele segregation type among the six accessions, as expected, and 100 InDels showed a heterozygous genotype in at least one of the six accessions. A total of 171 (74%) InDels from 231 InDels showed polymorphic bands between the two native accessions of Korea, M05, and M06.

### 2.4. Cluster Analysis and Fingerprinting of the Six Selected Accessions

The phylogenetic relationship was constructed with dendrogram coefficients using the numerical taxonomy system of multivariate programs (NTSYS) cluster analysis (Figure 3). An unweighted pair group method with an arithmetic mean (UPGMA) dendrogram was constructed for the six milk thistle accessions based on the 177 InDel polymorphisms, in which the Jaccard’s similarity coefficients ranged from 0.36 (M05 vs. M06) to 0.61 (M01 vs. M03). A pair of M01 and M03 were estimated as having the highest genetic similarity. Two accessions native to Korea (M05 and M06) were clustered separately from the four accessions (M01–M04) that originated from Canada, Germany, North Korea, and Moldova, respectively. For fingerprinting of the six selected accessions, a minimum marker set was developed using the InDel markers system. A set of six InDels (SM034, SM026, SM102, SM135, SM182, and SM176) were screened out of 177 primers based on sharp, clear, and reproducible bands (Figure 4A,B), which completely discriminated all six accessions. 

## 3. Discussion

Milk thistle (*Silybum marianum* L. Gaertn.) is a medicinal plant that contains silymarin, a compound that is beneficial for people with chronic liver disease [1]. It is native to the Mediterranean Basin and was introduced as a crop in Europe, North and South America, Asia, and Southern Australia [10]. The leaves and achenes of milk thistle are used as raw materials for food in Korea. Milk thistle extract is also available as an over-the-counter (OTC) medication in Korea (Ministry of Food and Drug Safety, 2022). Currently, Korean pharmaceutical companies import all the milk thistle extracts they use. This reliance on imports exposes them to risks such as supply chain disruptions. By localizing and standardizing the production of milk thistle, Korean pharmaceutical companies can reduce their reliance on imports and mitigate these risks. Localizing production means that milk thistle will be grown and processed in Korea. Standardizing production means that there will be a consistent quality of milk thistle available in Korea. This will ensure that Korean pharmaceutical companies have a secure supply of milk thistle, even in the event of supply chain disruptions or other challenges. 

Breeding high-quality milk thistles can be achieved through understanding the phenotypic variation and genetic diversity within the population. There are several challenges in milk thistle breeding that obstruct the cultivation and industrialization of milk thistle, such as spiny leaves, fruit dispersion, asynchronous flowering, unstable yield quality, and crop stability [4,31]. In this study, we investigated six traits associated with the breeding goals for milk thistle, over a three-year period (Appendix A). In the phenotypic evaluation among the selected six accessions, the weight of 100 achenes was increased as the number of flower heads and involucre diameter increased. This is consistent with previous studies concluding that the number of seeds per plant increased, ranging from a 484 to 1359 head per plant increase [32]. Both Korean native accessions, M05 and M06, were morphologically and genetically distinctive from the other four accessions. In M05, both the weight of 100 achenes and the number of flower heads were the highest (Table 1). Shim et al. (2020) reported that M05 contained the highest contents of silybin B in dried achenes (3.50 mg/g) among the six accessions, which is one of the major active constituents of silymarin [33,34]. Thus, M05 is a promising breeding material for milk thistle production in Korea. It has a number of desirable traits that make it well-suited to commercial production, including high yield, high silybin B content, and adaptability to a wide range of Korean climates and environments. Meanwhile, M06 showed a small spine length and small involucre diameter, along with the highest number of branches (Figure 2, Table 1). A particularly long spine length reduces work efficiency during cultivation and harvesting. The short spine length of M06 is considered a valuable trait for achieving optimal breeding target traits. M06 was genetically and morphologically distinct from other materials following phylogenetic analysis (Figure 3 and Appendix A). Overall, M06 has the potential to benefit commercial cultivation and breeding programs to reduce spine length. 

Molecular breeding for milk thistle, as a non-model medicinal plant, is far behind that of model crops, as there are insufficient genome data and efficient molecular markers. A few studies on *Silybum marianum* employing RAPD, AFLP, DArT array, and ISSR primer systems were conducted [4,35,36,37]. Marker-assisted selection (MAS) is a breeding technique that uses molecular markers to identify and select for desired traits in plants [35]. MAS can improve the productivity and accuracy of classical plant breeding by reducing its time consumption. InDels and SNPs are the most widely used PCR-based marker systems in MAS [38]. InDels are derived from the insertion of retrotransposons or other mobile elements, unequal crossover events, or slippage in simple sequence replication [39]. It is known that mutations in DNA repair genes are also related to the occurrence of indels [40]. They have a low frequency of homoplasy, indicating that there is an adequately low probability of two InDel mutations of exactly the same length occurring at the same genomic position [19]. InDels in genes with functional diversity between alleles are highly useful for marker-assisted selection or QTL mapping [41,42]. 

We conducted the first large-scale study of genome-wide InDel development in milk thistle (Table 3). Six milk thistle accessions, including two native Korean lines, were sequenced. After the strict filtering of heterozygotes for the reference allele, 30,845 InDels remained from the 238,988 InDel variants. Of the ‘303’ type, 231 InDels were selected, expected to represent the most polymorphic allele type, and 177 InDels (77%) of comparatively high polymorphism percentages were observed among the six selected validation accessions (Appendix A). These InDel primers were expected to serve as a high-potential tool for genetic discrimination among the *Silybum marianum* species. A novel marker set of 177 InDels with high amplification rates and high polymorphism can be utilized for genetic studies, such as pedigree analysis and seed purity test of a parental line and F1 hybrid (Figure 4). In allele types, there was a discrepancy in InDel sizes between the predicted and actual size. This discrepancy could be due to alignment to the primitive reference genome at the assembled contig level. 

Phylogenetic analysis using 177 InDels clarified the relationship among the six genotypes (Figure 3). Genetically diverse parents have the possibility to generate higher heterosis within phenotypes [43]. In milk thistle, ecotypes of different geographical regions with various ratios of flavonolignan compounds constituted a gene pool for plant improvement [32]. There was high variability in silymarin content and composition among natural populations in Iran, Egypt, and Italy. Low variability among non-native populations in New Zealand was reported [44]. Several milk thistle varieties and lines have been developed and registered in Poland, Hungary, Germany, England, and New Zealand [32,45]. In order to breed useful milk thistle cultivars, it is necessary to investigate the agronomical traits, genetic diversity, and silymarin contents of the breeding materials.

Orphan crops are often native to a particular region and have been cultivated for centuries, but they have not received the same level of research and development as other popular crops [8]. Their germplasm collections are not complete, and lack a full understanding of the genetic diversity of the plants. Modern genetic and genomic tools can be used to improve crop breeding, which could help ensure global food and nutritional security. In this study, we conducted a phenotypic evaluation of 220 milk thistle plants and developed the first large-scale InDel markers, which may serve as a foundation for breeding programs and genetic studies, including pedigree analysis, origin and evolutionary analysis, population structure and diversity analysis, QTL mapping, and marker-assisted selection.

## 4. Materials and Methods

### 4.1. Plant Materials

For the selection and standardization of domestic milk thistle resources, we collected 172 milk thistle plant seeds from local markets across Korea (Appendix A). In addition, six accessions were distributed from the National Agrobiodiversity Center, National Institute of Agricultural Sciences, Rural Development of Administration (RDA), Korea. For phenotypic evaluation, a total of 220 accessions, composed of 172 accessions from local markets and 48 accessions isolated from 6 accessions, were grown in Hwaseong, Gyeonggi-do, Korea, in 2018 and 2019. Of them, six representative accessions were selected based on morphological characteristics. Four accessions (M01–M04) among the six selected accessions were distributed from RDA, originating from Canada, Germany, North Korea, and Moldova (Table 1). The two other accessions, M05 and M06, were collected from local markets in Korea. The six selected accessions were advanced by self-pollination and used for the resequencing analysis and phenotypic evaluation in 2019 to 2021.

### 4.2. Phenotypic Evaluation

To select a representative plant for consideration of its morphological characteristics and industrial value, six phenotypic traits were evaluated for 220 milk thistle accessions in 2018 and 2019, i.e., plant height (cm), seed weight (g), number of flower heads, seed weight(g)/ number of flower heads, spine length (mm), and plant color at harvest. Following the selection of six representatives, we evaluated six quantitative characteristics from 2019 to 2021, i.e., plant height (cm), number of branches, the diameter of involucre with spine tips (mm), number of flower heads, 100-achene weight (g), and spine length (mm). Statistical analyses were conducted to determine significant differences in six agronomic traits among six accessions, using analysis of variance (ANOVA) and Duncan’s multiple range test in the R package. Frequency distributions were calculated for each trait using Microsoft Office Excel 2016. We used the statistical analysis package SPSS 12.0KO for Windows to calculate Pearson’s correlation coefficients, principal component analysis (PCA), and hierarchical cluster analysis using Ward’s method. Six agriculturally superior plants were selected that represented the phenotypic variation of traits and had the potential to be used as breeding materials from a breeder’s perspective.

### 4.3. Genome Resequencing and Assembly

DNA from a single plant of each accession was extracted using the Cetyl Trimethyl Ammonium Bromide (CTAB) method [46]. Each DNA was quantified by NanoDrop 2000 (Thermo Fisher Scientific, Waltham, MA, USA.), and only the high-quality DNA samples for genome sequencing were used. For the first step, high-depth resequencing was conducted on six milk thistle accessions to identify InDel markers. An Illumina paired-end (P.E.) library with a 400 bp insert size was constructed according to the manufacturer’s recommendations, and the library was sequenced on Illumina Novaseq with 2 × 150 bp. Low-quality sequences (Phred score ≤ 20) and Illumina adapter sequences were removed in raw fastq files using Trimmomatic v.0.39 (http://www.usadellab.org/cms/?page=trimmomatic (accessed on 1st September 2021)). The reference genome sequence of *Silybum marianum* was downloaded from the NCBI database (Genbank assembly acc# GCA_001541825.1). The trimmed data were aligned to the reference genome using BWA-MEM (version 0.7.1.7). 

### 4.4. Variant Calling and InDel Screening 

The alignment data were transformed into a binary alignment map (BAM) format via SAMTools [47]. ‘Mark duplicates’ in the Picard tool (Broad Institute, Boston, MA, USA) were applied to remove replicate reads. To reduce the inaccurate alignments, a GATK-HaplotypeCaller was used to conduct local realignment around the insertions and deletions, read base quality recalibration, and variant calling [48]. The dataset was further filtered using two approaches: (i) relaxed filtering for minDP ≥ 10, minGQ ≥ 30, missing data ≤ 20% for each locus, and removing monomorphic variants among six milk thistle accessions; and (ii) stricter filtering for heterozygote and missing data. Then, sample genotypes were grouped according to modified allele frequencies, which are coded as a three-digit number. Genotype groups were represented as ‘105’, ‘204’, ‘303’, ‘402’, and ‘501’ (Appendix A). The first digit represents the number of accessions with homozygotes for the reference allele among the six accessions, the second digit represents the number of accessions with the heterozygote allele, and the third digit represents the number of accessions for the alternative allele, respectively. Of the ‘303’ group, 234 InDels were selected and PCR primers were designed from the flanking region using Primer3 [49]. Of them, 231 InDels with size differences between 15 bp and 100 bp were arbitrarily chosen for experimental resolvability under agarose gel electrophoresis.

### 4.5. PCR Amplification

For validating the identified InDels, PCR was performed on a SimpliAmp thermocycler (Thermo Scientific, Waltham, MA, USA) in a 20 μL reaction volume containing 50 ng of DNA template and 0.5 μL of each forward and reverse primer, making a total of 10 μL (Bioneer, Daejeon, Republic of Korea), 2 μL of 10 × buffer, 0.5 μL of 2.5 mM dNTPs, and 0.1 μL of Taq polymerase (IN5001-0500; Inclone, Yongin, Republic of Korea), under the following conditions: initial denaturation at 94 °C for 4 min, followed by 35 cycles of denaturation at 94 °C for 1 min, annealing at 55 °C for 30–60 s, and extension at 72 °C for 30–60 s, with a final extension at 72 °C for 7 min. The PCR products were electrophoresed (BioFACT, Daejeon, Republic of Korea) on 1% or 2% agarose gel at 100–160 V for 20–40 min in a 0.5 × TBE buffer. Gels were visualized using a gel imager (Korea Lab Tech, Seongnam, Republic of Korea). To consider the resolution of agarose gel, a single PCR product with a fragment ≥ 30 bp larger than the reference and alternative alleles was determined to comprise C and D alleles, according to the order of detection of the six selected accessions (M01–M06).

### 4.6. Genetic Diversity Assay 

To generate molecular data matrices, the presence or absence of clear bands was scored in every accession for each primer pair, and recorded either as 1 (presence of a fragment) or 0 (absence of a fragment). A phylogenetic tree was conducted based on genetic distances and the unweighted pair group method with an arithmetic mean (UPGMA) using the Jaccard functionality of NTSYSpc Version 2.21 m (Exeter Software, Setauket, NY, USA).

## 5. Conclusions

Milk thistle (*Silybum marianum* L. Gaertn) is an important medicinal plant for chronic liver disease. We selected six accessions that showed significant differences in agricultural characteristics to be used as initial materials for genomic research into milk thistle. A large-scale development of genome-wide InDels using resequencing analysis was performed. As a result, 177 InDels with reliable polymorphisms were developed from the resequencing data of the six selected milk thistle accessions. These plant materials and InDels could be valuable resources for the identification, conservation, breeding programs, and industrialization of *Silybum marianum*.

## Figures and Tables

**Figure 1 plants-12-02702-f001:**
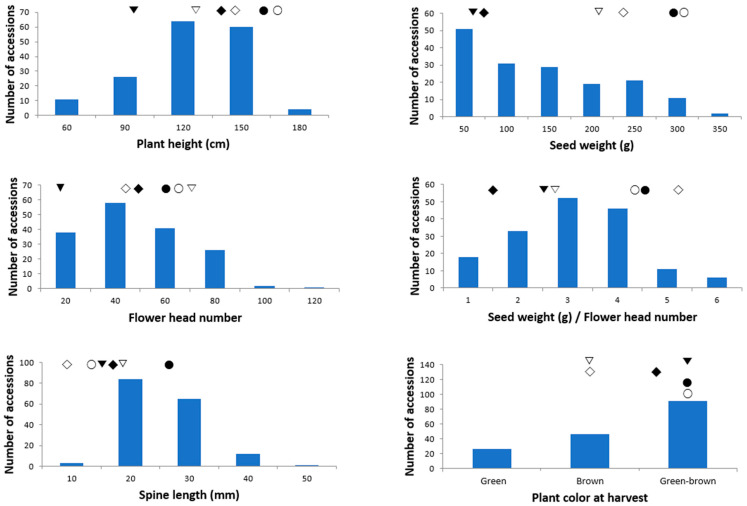
Frequency distribution for each trait of the 220 milk thistle accessions. Six representative milk thistle accessions are displayed as figures on the graph. M01 (▼); M02 (▽); M03 (◆); M04 (◇); M05 (●); and M06 (○). Plant color at harvest was classified into three categories based on color charts: green, brown, and green-brown.

**Figure 2 plants-12-02702-f002:**
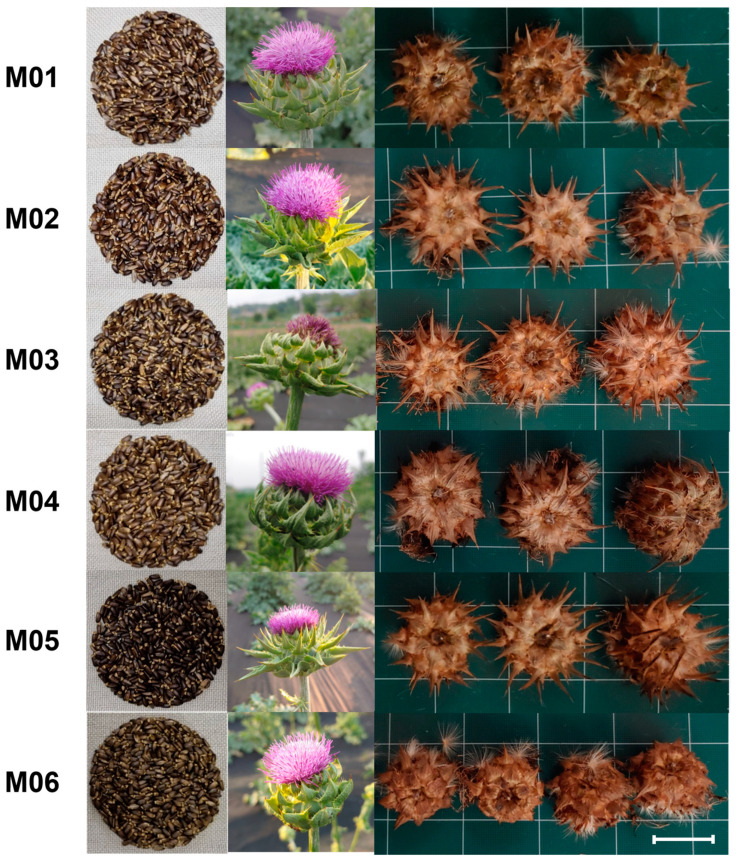
Morphological characteristics of the selected six milk thistle accessions: (from left) seed, flower, and dried involucre after harvest. The scale bar indicates 40 mm.

**Figure 3 plants-12-02702-f003:**
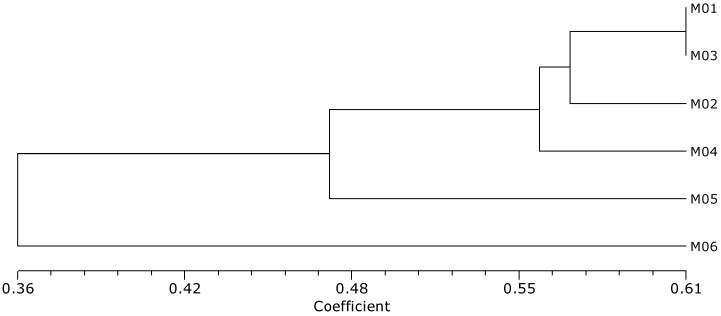
The phylogenetic relationship among the six selected accessions of milk thistle was constructed using NTSYS cluster analysis; a dendrogram showing the genetic similarity based on the 177 InDel polymorphisms.

**Figure 4 plants-12-02702-f004:**
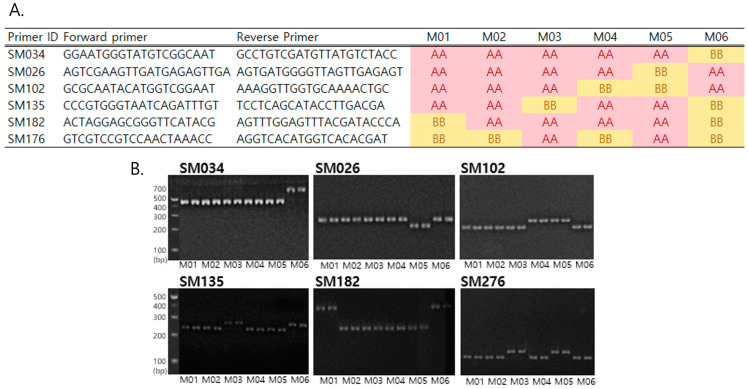
Minimum InDel marker set for fingerprinting the six selected accessions across six markers. (**A**) Primer list including primer ID, primer sequences, and allele types of six accessions. Homozygous reference alleles (AA) are shown in red, and homozygous alternative alleles (BB) in yellow. (**B**) Agarose gel images were analyzed via SM034, SM026, SM102, SM135, SM182, and SM176 from two samples of each accession, respectively.

**Table 1 plants-12-02702-t001:** Comparison of the phenotypic variations in the 6 accessions between M01–06 and 220 accessions (accs.). Six agronomical traits were evaluated: plant height (PH, cm), seed weight (SW, g), flower head number (FHN), SW/FHN, spine length (SL, mm), and plant color at harvest (PC). PC was classified into three categories based on color charts: green (1), brown (2), or green-brown (3). The mean ± SD are shown in a separate column.

Variable	M01	M02	M03	M04	M05	M06	Mean		Contrast
							M01–M06	220 Accs.	M01–06:220 Accs.
PH	121.5	100.1	136.2	141.6	131.9	134.2	127.6 ± 15.0	109.8 ± 27.0	*
SW	116.1	79.5	118.8	97.2	142.0	109.1	110.4 ± 21.1	112.2 ± 85.0	NS
FHN	40.4	38.7	37.7	25.4	43.9	45.0	38.5 ± 7.0	37.5 ± 21.7	NS
SW/FHN	2.7	2.1	3.0	3.8	3.2	2.4	2.9 ± 0.6	2.6 ± 1.4	*
SL	39.7	46.4	43.9	28.7	45.8	22.9	37.9 ± 9.8	20.7 ± 6.0	*
PC	2.7	2.7	2.7	2.3	2.7	2.7	2.6 ± 0.1	2.6 ± 0.8	NS

* Significant at *p* < 0.05, NS, not significant.

**Table 2 plants-12-02702-t002:** Results of the first two axes (PC1, PC2) of the principal component analysis of morphological traits within the relationships between 220 milk thistle accessions. Six agronomical traits were evaluated: plant height (PH, cm), seed weight (SW, g), flower head number (FHN), SW/FHN, spine length (SL, mm), and plant color at harvest (PC). PC was estimated as one of three types according to color charts: green (1), brown (2), or green-brown (3).

	PC1	PC2
Eigenvalue	3.37	1.06
Variance (%)	56.2	17.6
Cumulative variance (%)	56.2	73.8
Variable		
SW	**0.97**	−0.30
FHN	**0.91**	−0.14
PH	**0.86**	−0.44
SW/FHN	**0.77**	−0.43
SL	−0.23	**0.84**
PC	0.33	**−0.72**

**Table 3 plants-12-02702-t003:** Percentage of reads in each accession mapping to the reference genomes of *Silybum marianum* L. Gaertn.

Accession	Total Trimmed	Mapped	Unmapped	Properly Paired		Properly Paired (%)	Mapped (%)	Unmapped (%)	Properly Paired Mapped (%)
M01	108,554,670	97,690,066	10,864,604	81,978,182		75.5	90.0	10.0	83.9
M02	136,302,598	116,215,035	20,087,563	96,982,188		71.1	85.3	14.7	83.5
M03	146,110,578	131,474,969	14,635,609	109,901,174		75.2	90.0	10.0	83.6
M04	146,000,972	130,085,152	15,915,820	108,315,504		74.2	89.1	10.9	83.3
M05	121,009,526	104,304,345	16,705,181	86,915,402		71.8	86.2	13.8	83.3
M06	122,017,832	107,898,658	14,119,174	90,503,310		74.2	88.4	11.6	83.9
Total	779,996,176	687,668,225	92,327,951	574,597,760	Mean	73.7	88.2	11.8	83.6

**Table 4 plants-12-02702-t004:** Polymorphic types among six accessions were grouped according to modified allele frequency, which is represented by a three-digit number. The first number of three digits represents the number of accessions with homozygotes for the reference allele among six accessions, the second digit represents the number of accessions with a heterozygote allele, and the third digit represents the number of accessions with homozygotes for the alternative allele, respectively. Distribution of various sizes of InDel polymorphism (more than 15 bp, more than 20 bp, and more than 25 bp) was identified among the six selected accessions of milk thistle.

Polymorphic Types	InDel	Size of InDel Polymorphism		
		≥15 bp	≥20 bp	≥25 bp
‘105’	4750	259	165	114
‘204’	3334	207	127	92
‘303’	3410	234	161	115
‘402’	6114	480	321	221
‘501’	13,237	1161	777	557
Total	30,845	2341	1551	1099

## Data Availability

All data that support the findings within this study are available at NCBI GenBank with the following accession numbers: PRJNA887484. This includes genomic sequencing data for the selected six accessions (M01–M06).

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
