# Peer review of "A Genomic Evaluation of Six Selected Inbred Lines of the Naturalized Plants of Milk Thistle (Silybum marianum L. Gaertn.) in Korea"

_plants, 2023, doi:10.3390/plants12142702_

Round 1

Reviewer 1 Report (New Reviewer)

1. Change the word 'some' by 'six' in the title;

2. Line 115, change 'plants'  to 'accessions' , and the same in Figure 1;

3. Line 113,  Silybum marianum need to write in S. marianum;

4. Figure 4B, the quality and information need to be improved;

Author Response

[Response to Reviewer #1]

Comment 1) Change the word 'some' by 'six' in the title.

Response: Thank you for your careful reading of our manuscript. We have changed the word 'some' by 'six' in the title.

Comment 2) Line 115, change 'plants' to 'accessions' , and the same in Figure 1;

Response: We thank the reviewer for pointing this out. We have changed 'plants' to 'accessions' in lines 119, Figure 1, and the same in lines 31, 120, 125, 126, 129, 136, 144, 341, 342, 352, Table 1, Table 2, and Table S1.

Comment 3) Line 113, Silybum marianum need to write in S. marianum.

Response: Thank you for your careful reading of our manuscript. We have changed ' Silybum marianum ' to ' S. marianum ' in lines 117.

Comment 4) Figure 4B, the quality and information need to be improved.
Response: According to the reviewer’s comments, we improved the quality of the pictures and added molecular markers in Figure 4B.

Reviewer 2 Report (New Reviewer)

The biggest problem for this study is experimental design.

(1) The authors didn't clearly explain the reason why they chose those 6 accessions for the study, and they didn't give any supportive evidence for their choice.

(2) Only 6 accessions were chosen as representatives of the whole population were not good enough.

(3) The analysis and results are too thin to support the conclusions. More deeper analyses are needed. 

Author Response

[Response to Reviewer #2]

Comment 1) The authors didn't clearly explain the reason why they chose those 6 accessions for the study, and they didn't give any supportive evidence for their choice.
Response: Thank you for your careful reading of our manuscript. We selected six representatives (M01-M06) from 220 plants of milk thistle based on morphological traits. To select representatives of 220 plants, six agronomic traits were evaluated for two years, and conducted histogram analysis, phenotypic variation analysis, and Pearson correlation analysis. In addition, we conducted principal components analysis (PCA) and phylogenetic tree analysis using all the phenotypes (Table 2). And cluster analysis of milk thistle plants based on six agronomic traits resulted in three main groups by morphological affinity (Figure S1). And we colored six representative milk thistle plants in red. Based on the results, we selected six agriculturally superior plants that represent the phenotypic variation of traits and have the potential to be used as breeding materials from a breeder's perspective. The selected plants, including two Korean natives (M05 and 06), were used as initial materials for genomic research, such as DNA marker development in this study, and also utilized for breeding material. Now we have completed the sequencing of the entire genome of M05 (a Korean-native plant with a high silymarin content) and a comparative genomic analysis of the genomes of other Asteraceae species. This result will also be submitted for publication soon.

Comment 2) Only 6 accessions were chosen as representatives of the whole population were not good enough.
Response: Thank you for your careful reading of our manuscript. In this study, we used only 6 accessions, but we want to select additional representatives from the remaining accessions using the InDel markers developed here in the near future.

Comment 3) The analysis and results are too thin to support the conclusions. More deeper analyses are needed. 
Response: Thank you for your careful reading of our manuscript. We added principal components analysis (PCA) and phylogenetic tree analysis using all the phenotypes (Table 2). Cluster analysis of milk thistle plants based on six agronomic traits resulted in three main groups by morphological affinity (Figure S1). In addition, we added this point to the discussion in line 325.

Round 2

Reviewer 2 Report (New Reviewer)

For the selection fo these 6 accessions, my suggestion is to move Fig. S1 from supplementary file to the main text. However, the presenting format of this figure needs to be changed and improved. This figure is important as a supportive evidence to show the reason why you chose these accessions for your study.

Author Response

[Response to Reviewer #2]

Comment 1) For the selection fo these 6 accessions, my suggestion is to move Fig. S1 from supplementary file to the main text. However, the presenting format of this figure needs to be changed and improved. This figure is important as a supportive evidence to show the reason why you chose these accessions for your study.

Response: We thank the reviewer for pointing this out. We improved the quality of Figure S1 to make the text clearer and the clusters more distinguishable. This figure is important supporting evidence, but it is too large to be included in the main text. Therefore, I would like to have it placed in Figure S1. I hope you will understand.

This manuscript is a resubmission of an earlier submission. The following is a list of the peer review reports and author responses from that submission.

Round 1

Reviewer 1 Report

Dear author,

  Milk thistle (Silybum marianum) is very important for its medicinal usage for chronic liver disease. The manuscript collected seven traits of 220 plants. However, the manuscript does not display the data sufficiently. The manuscript should do genetic analysis using all the traits, such as PCA, structure, phylogenetic tree. In addition, the manuscript did not show how to select the representatives. Six representatives were resequenced, and Indels were identified using GATK in manuscript. Indels with long bases detected using GATK were not accurate. The validated indels should apply in more samples rather than the six sample. In a word, the author should do more analysis with those data and found more interesting found.

Lines 92-93: Last sentence could not infer that “InDels are better phylogenetic markers for tracing the contributions of the ancestral species than SNPs and SSRs”, please provide more evidence.

Line 115: 32cm to “32 cm”.

Figure S2: Why all the traits were highly correlated with so low correlated value in the Figure.

The author need calculate pearson correlation coefficient, and compare with the result in Figure 2.

Figure 1: How to select the representatives?

Line 112: Please do principal components analysis and phylogenetic tree analysis using all the phenotypes, and color six representative milk thistle plants in red.

Line 136-137: Please do variance analysis using all the phenotypes and the representatives.

Table 1 and Table 2 could not fully display the data. If the author could demonstrate how to select the six representatives, Table 2 cloud take as a supplement.

Line 150: 120.5GB to ‘120.5 GB’.

Line 170-171: What does the modified allele frequency mean?

Line 175-176: 39bp, 100bp, and so on.

Line 172: Please explain ‘303’.

Figure S1. The legend is not clear.

Table 4: What about polymorphic types?

Figure 3 should delete.

Reviewer 2 Report

Jeehyoung Shim et al. в работе "An application of the genomic evaluation to some selected inbred lines of the naturalized plants of milk thistle (Silybum marianum L. Gaertn.) in Korea"  developed a PCR marker system for genotyping milk thistle plants. The novelty of the study is due to the fact that similar studies have not yet been carried out for milk thistle. To create a system of genetic markers, a team of researchers conducted a pheynotypic analysis of more than 200 plants from different locations. Then sequencing of genomes of 6 milk thistle variations was carried out. Next, the authors conducted a detailed search for insertions and deletions, which, after two-stage filtration, served as the basis for constructing a system of PCR markers. This work is quite relevant and important for agriculture and biodiversity research.

Reviewer 3 Report

This manuscript reports on the selected plant materials and markers for analyzing the cultivar identification, genetic diversity of breeding materials, and ecological studies of native to Korea. 

Overall, the authors have made a good attempt at adding value to the genomic evaluation of milk thistle (Silybum marianum L. Gaertn.) in Korea.

I found it a little difficult to decide what to make of this work.

major comments

The authors conclude that six representative accessions were selected based on morphological characteristics and homozygosity within accessions, but I cannot observe that in this manuscript.

In my opinion, the authors need to describe the passport data of 200 plants.

The data presented in this study are rather preliminary and do not support the importance of the selected plant materials and markers.

minor comments

Figure 2

There is no scale in Figure 2.

Table 2

What does the "abcd" indicate?